# Determination of the Bacterial Community of Mustard Pickle Products and Their Microbial and Chemical Qualities

**DOI:** 10.3390/biology12020258

**Published:** 2023-02-06

**Authors:** Hung-I Chien, Yu-Fan Yen, Yi-Chen Lee, Pi-Chen Wei, Chun-Yung Huang, Chih-Hua Tseng, Feng-Lin Yen, Yung-Hsiang Tsai

**Affiliations:** 1Department of Seafood Science, National Kaohsiung University of Science and Technology, Kaohsiung 811213, Taiwan; 2Department of Bioscience and Biotechnology, National Taiwan Ocean University, Keelung 202301, Taiwan; 3Department of Fragrance and Cosmetic Science, Kaohsiung Medical University, Kaohsiung 807378, Taiwan

**Keywords:** mustard pickle, sulfite content, high-throughput sequencing, microbiome, quality

## Abstract

**Simple Summary:**

The sulfite contents from bleach in all commercially available mustard pickle products exceeded the allowable limit of food additives (30 ppm), with a failure rate of 100%. Although the samples contained no food pathogens, the high-throughput sequencing results revealed potential environmental contamination in samples.

**Abstract:**

We assessed the microbial and chemical qualities and microbiomes of 14 mustard pickle products coded sequentially from A to N and sold in traditional Taiwanese markets. The results showed that the aerobic plate count and lactic acid bacteria count of commercially available mustard pickle products were 2.18–4.01 and <1.0–3.77 log CFU/g, respectively. Moreover, no coliform bacteria, *Escherichia coli*, *Staphylococcus aureus*, *Salmonella* spp., or *Listeria monocytogenes* were detected in any of the samples. Analysis of the chemical quality showed that the sulfite content of all samples exceeded 30 ppm, which is the food additive limit in Taiwan. Furthermore, the mean contents of eight biogenic amines in the mustard pickle product samples were below 48.0 mg/kg. The results of high-throughput sequencing showed that the dominant bacterial genera in sample A were *Proteus* spp. (25%), *Vibrio* (25%), and *Psychrobacter* (10%), in sample C they were *Weissella* (62%) and *Lactobacillus* (15%), in sample E it was *Lactobacillus* (97%), and in sample J it was *Companilactobacillus* (57%). Mustard pickle product samples from different sources contained different microbiomes. The dominant bacterial family was *Lactobacillaceae* in all samples except for sample A. In contrast, the microbiome of sample A mainly consisted of *Morganellaceae* and *Vibrionaceae*, which may have resulted from environmental contamination during storage and sales. The result of this work suggests it may be necessary to monitor sulfite levels and potential sources of bacterial contamination in mustard pickle products, and to take appropriate measures to rule out any public health risks.

## 1. Introduction

Mustard pickle (*Brassica juncea*) is a fermented vegetable with a sour taste and is the most common traditional fermented food in Taiwan [1]. Mustard pickle is produced by drying harvested mustard in the sun and then placing it in a bucket in layers, with salt (NaCl) added to the surface of each layer at approximately 13% of the mustard weight [2]. Finally, the top layer is covered with a heavy object (as shown in Figure 1). Mustard pickles are processed after a fermentation time of 4–5 months. The samples had the outer leaves removed, were immersed in sulfurous acid solution for bleaching, and then drained, providing the finished product [1,2]. In traditional Taiwanese markets, the products are typically stored at room temperature for sale [2].

The manufacture of most fermented vegetable products, including kimchi in South Korea, sour Chinese cabbage in China, and sauerkraut in Europe, is highly dependent on naturally occurring lactic acid bacteria (LAB), such as *Lactobacillus*, *Weissella*, *Leuconostoc*, and *Pediococcus*, for fermentation [3,4,5]. Because many microorganisms cannot grow in culture medium, studies in which traditional medium culture is used cannot reveal the actual microbiome in food [6,7]. In addition, many molecular biological techniques, such as real-time polymerase chain reaction (real-time PCR), denaturing gradient gel electrophoresis, and terminal restriction fragment length polymorphism do not comprehensively reflect the types and proportions of microorganisms in specific foods [8]. In contrast, high-throughput sequencing (HTS) technology could correctly determine the microbial flora in foods [9]. The Pacific Biosciences (PacBio) single-molecule real-time (SMRT) sequencing platform is a new third-generation high-throughput sequencing technology that uses a different sequencing method to generate long read sequences of base pair data (10–15 kb on average using P6–C4 chemistry) without any GC bias being introduced [10,11]. Its read length is between 3000 and 15,000 bp with an accuracy rate of 99% [10,11]. This means that SMRT sequencing technology is an excellent tool for generating full sequence data of 16S rRNA genes to identify bacterial diversity and community structure at a species level, enabling it to be used to reveal differences in bacterial samples between related samples [12,13].

Kung et al. [2] investigated the safety of 37 mustard pickle products available on the market and found that 82% of the samples contained sulfites at levels exceeding the standards for food additives in Taiwan (30 ppm). It was also found that the hygienic quality of mustard pickle sold in Taiwan was previously unfavorable [2]. There have also been news reports that processing factories for “pickle beef instant noodles”, which are well-known in China, were fined by hygiene departments because the pickle manufacturing environment was unsanitary and good hygienic practice of food preparation was not followed during on-site production [13]. The current study was conducted to investigate whether the microbial and chemical-related hygienic qualities of commercially available mustard pickle products have improved in the past ten years, compared with the reports of Kung et al. [2]. In addition, full-length sequencing of the bacterial 16S rRNA gene in the samples was performed using PacBio SMRT to analyze the bacterial community in mustard pickle products. Therefore, the novelty of this study is that it is the first to use high-throughput sequencing technology to study the bacterial community of mustard pickles in Taiwan.

## 2. Materials and Methods

### 2.1. Sample Collection

From January 2021 to March 2021, mustard pickle products coded in order from A to N were purchased from 14 different traditional markets in Taiwan. Three samples (triplicates) were analyzed for each traditional market. All samples were sold at room temperature (25–27 ℃), placed in crushed ice immediately after collection, and transported to the laboratory for analysis within 2–3 h.

### 2.2. Microbial Assay

To determine the aerobic plate count (APC), 10 g of the sample was placed in a sterilized blender bottle containing 90 mL of saline and homogenized with a homogenizer (Osterizer, Brampton, ON, Canada) for 2 min to obtain a 10x dilution solution. The homogeneous solution (1.0 mL) was added to a sterilized test tube containing 9 mL of sterile saline to prepare 10^2^-, 10^3^-, and 10^4^-fold diluted solutions in sequence. These solutions (0.1 mL) were applied in duplicate to trypticase soy agar (Difco, BD, Sparks, MD, USA) and the petri dishes were placed in a 30 ℃ incubator for cultivation. The number of colonies was counted after 48 h [14]. In addition, 1.0 mL of the diluted solutions (in duplicate) was obtained from the APC solutions with different dilution ratios and poured into de Man, Rogosa, and Sharpe agar (MRS agar) for incubation at 30 °C for 24–48 h. The number of colonies was considered as the number of LAB [14]. A 3M Petrifilm *E. coli*/Coliform Count Plate and 3M Petrifilm Yeast and Mold count plate (3M Microbiology, St. Paul, MN, USA) were used to detect coliform and *E. coli* as well as yeast and mold, respectively [15]. The analyses were performed according to the manufacturer’s instructions. Furthermore, CHROMagar^TM^ *Staph aureus*, CHROMagar^TM^ *Salmonella* Plus, and CHROMagar^TM^ *Listeria* (CHROMagar, Paris, France) were used to determine the bacterial counts of *Staphylococcus aureus*, *Salmonella* spp., and *Listeria monocytogenes*, respectively [15]. The procedures and culture on CHROMagars were performed according to the manufacturer’s instructions. Briefly, 0.1 mL of the APC homogeneous solution or diluted solution was spread onto the CHROMagar medium of various pathogenic bacteria, and then the growth of the colony after culture was observed. However, in the detection of *L. monocytogenes*, 0.1 mL of APC homogeneous solution or diluted solution was inoculated into Fraser broth at 35 °C for 24 h for the enrichment step. If the Fraser broth turned black, 0.1 mL of the enrichment solution was spread onto CHROMagar^TM^
*Listeria* for determination [15].

### 2.3. Chemical Composition Analysis

To determine water activity, the crushed samples (3 g) were placed in a plastic vessel to determine the water activity using a water activity meter (LabSwift-aw, Novasina, Lachen, Switzerland). To determine the moisture content, approximately 5 g of finely crushed samples was placed in an aluminum dish, and the moisture content of the sample was calculated using an infrared drying moisture meter when the sample reached a constant weight after heating at 105 ℃. To determine the sample pH, 10 g of the sample was homogenized with distilled water (90 mL) with a homogenizer (Polytron PT3100D, Kinematica, Littau-Luzern, Switzerland) for 120 s and filtered. The filtrate of samples was measured using a HORIBA pH meter F-71S (Kyoto, Japan). Precipitation titration (Mohr’s method) was performed to determine the salt content [16]. In terms of titratable acid content, 10 g of sample was homogenized with 90 mL of distilled water and filtered. The filtrate was titrated with 0.1 N NaOH to pH 8.1 as the end point, and lactic acid was used as the standard solution to calculate the titratable acidity [17]. Total volatile basic nitrogen (TVBN) in samples was determined using on the Conway’s dish microdiffusion method proposed by Cobb et al. [18]. Sulfites in the mustard pickle samples were extracted by aeration distillation, and the sulfite content was calculated by titrating the samples with 0.01 N NaOH [19]. The method described by Chou et al. [20] was used as a reference to determine the nitrite content using high-performance liquid chromatography (HPLC). Biogenic amines in mustard pickle products were extracted with trichloroacetic acid and derivatized with dansyl chloride before high-performance liquid chromatography analysis. The contents of eight biogenic amines in the samples were determined as described by Chen et al. [21].

### 2.4. High-Throughput Sequencing Method

The 100 g of crushed samples was covered with gauze in a sterile environment and the watery sap was squeezed out. After the centrifugation (3500× *g*, 15 °C, 10 min) for liquid, and the supernatant was thrown away. After the precipitate was mixed with 9 mL of phosphate buffer solution, the solution was centrifuged again, and the supernatant was thrown away. The QIAamp PowerFecal DNA Kit (QP; Qiagen, Hilden, Germany) was used to extract the bacterial genomic DNA from the precipitate according to the manufacturer’s operating manual.

For SMRT sequencing of the full-length 16S rRNA gene, the primers of forward (5′-AGRGTTYGATYMTGGCTCAG-3′) and reverse (5′-RGYTACCTTGTTACGACTT-3′), containing a set of 16-nucleotide barcodes, were designed and used to perform PCR (2720 Thermal Cycler, Applied Biosystems, Foster City, CA, USA) to amplify the bacterial 16S rRNA gene. The PCR amplification program involved initial denaturation at 95 °C for 3 min, main denaturation step at 95 °C for 30 s, annealing at 55 °C for 30 s, and extension at 72 °C for 60 s, for 27 cycles. The final step involved heating at 72 °C for 5 min. The quality control of the amplified products was checked using 1% agarose gel electrophoresis and spectrophotometry (Nanodrop 1000, Thermo Fisher Scientific, Waltham, MA, USA). The results showed that the amplified products extracted from the samples were of poor quality, except for samples A, C, E, and J; thus, only these products were subjected to SMRT sequencing analysis.

The amplified products were sequenced using P6-C4 chemistry on a PacBio RS II (Pacific Biosciences, Menlo Park, CA, USA). Raw data were processed and refined using the quality clinical flow chart provided by Quantitative Insights into Microbial Ecology version 1.7 to ensure high accuracy in detecting the operational taxonomic units (OTUs). Representative sequences were identified by alignment of extracted high-quality sequences that showed 100% clustering of sequence identity. OTUs based on 97% threshold identity were selected using UCLUST software and the representative sequences were submitted to the RDP classifier to obtain classifications at the phylum, class, order, family, and genus levels.

### 2.5. Statistical Analysis

This study aims to understand whether there is an obvious relationship between the microbial counts (APC, LAB, and yeast and mold) and chemical characteristics (pH, moisture, water activity, titratable acidity, salinity, nitrites, sulfites, and TVBN) of commercial mustard pickle products. Therefore, Pearson’s correlation coefficient analysis was performed for the APC, LAB count, yeast and mold, pH, moisture, water activity, titratable acidity, salinity, nitrites, sulfites, and TVBN of commercially available mustard pickle samples. All data are shown as the mean of three samples, and SPSS version 22.0 software (SPSS, Inc., Chicago, IL, USA) was used to analyze data.

## 3. Results and Discussion

### 3.1. Microbiological Quality of Mustard Pickle Samples

Table 1 shows the APC, LAB, coliform, *Escherichia coli*, yeast and mold, and pathogenic bacteria (*S. aureus*, *Salmonella* spp., and *L. monocytogenes*) in the mustard pickle samples. The APC of the 14 commercially available mustard pickle samples was 2.18–4.01 log CFU/g (mean, 3.01 log CFU/g), LAB count was <1.0–3.77 log CFU/g (mean, 1.24 log CFU/g), and yeast and mold count was <1.0–4.85 log CFU/g (mean, 0.70 log CFU/g). No coliform, *E. coli*, or pathogenic bacteria (*S. aureus*, *Salmonella* spp., and *L. monocytogenes*) were detected in any of the samples. According to the *Sanitation Standard for Microorganisms in Foods* issued by the Taiwan Food and Drug Administration [22], the limit of *S. aureus* in mustard pickle is 10^2^ CFU/g, no *Salmonella* spp. should be detected, and the limit for *L. monocytogenes* is 10^2^ CFU/g. None of the three above food pathogenic bacteria were detected in the 14 commercially available mustard pickle samples. Therefore, the samples met the requirements of microbiological standards for foods. From the results of this study, it can be seen that the microbial numbers in samples sold in the market is low and no relevant food pathogens have been detected, which may be caused by the addition of sulfites in the mustard pickle production process [2]. Sulfite additives are intended primarily for controlling microbial growth, preventing browning and food spoilage in fermented food [23]. Therefore, the microorganisms in the sample were killed or inhibited.

### 3.2. Chemical Quality of Mustard Pickle Samples

Table 2 shows the pH, water activity, moisture content, salt content, titratable acidity, TVBN, and sulfite and nitrite levels in the samples. In the 14 mustard pickle samples, the pH was 3.39–4.16 (mean, 3.82), water activity was 0.914–0.960 (mean, 0.944), moisture content was 85.77–92.80% (mean, 89.70%), salinity was 3.02–8.55% (mean, 4.75%), titratable acidity was 0.51–1.66% (mean, 0.88%), TVBN was 16.6–36.0 mg/100 g (mean, 22.3 mg/100 g), sulfite content was 101.3–1065.4 ppm (mean, 440.5 ppm), and nitrite content was <0.01–0.65 ppm (mean, 0.32 ppm). According to the *Standards for Specification, Scope, Application and Limitation of Food Additives* published by the Taiwan Food and Drug Administration [24], the residual sulfite content in mustard pickle products should not exceed 30 ppm. Therefore, the sulfite residues of all commercially available mustard pickle samples (14/14, 100%) exceeded the limit. These findings are similar to those reported by Kung et al. [2], namely that 82% of the samples contained sulfites at levels exceeding 30 ppm. This result also suggests that sulfites were used for bleaching in the mustard pickle products and left high levels of residue, which may elicit allergic reactions such as asthma in consumers [25]. Therefore, it is a dangerous allergen causing health hazards.

Table 3 shows the contents of eight biogenic amines in the 14 mustard pickle samples, which were as follows: tryptamine 2.1 mg/kg, 2-phenylethylamine 9.1 mg/kg, putrescine 4.8 mg/kg, cadaverine 22.6 mg/kg, histamine 6.4 mg/kg, tyramine 48.0 mg/kg, spermidine 6.0 mg/kg, and spermine 0.3 mg/kg. Kung et al. [2] investigated the food safety of mustard pickle products sold in Taiwan and found high histamine contents of 89 and 74 mg/kg in two samples. However, in the current study, the histamine contents of all samples were below 17.3 mg/kg, showing low levels.

### 3.3. Correlation Analysis of Microbial and Chemical Qualities of Mustard Pickle Samples

Table 4 shows the Pearson correlation coefficients of the microbial and chemical qualities of the 14 mustard pickle samples. Significant positive correlations were observed between APC and LAB (r = 0.7913; *p* < 0.01), between water activity and the moisture content (r = 0.9020; *p* < 0.01), and between salt content and nitrites (r = 0.5487; *p* < 0.05). In addition, significant negative correlations were observed between pH and titratable acidity (r = −0.9390; *p* < 0.01), between water activity and salt content (r = −0.9384; *p* < 0.01), between water activity and TVBN (r = −0.5750; *p* < 0.05), between moisture and salt content (r = −0.8417; *p* < 0.01), and between moisture and TVBN (r = −0.5921; *p* < 0.05).

Compared with the results of Kung et al. [2], no significant differences were observed in the APC, coliform and *Escherichia coli*, pH, salt content, and titratable acidity between the previously described and current sample in the traditional market. Kung et al. [2] suggested that the histamine level of two commercial samples exceeded 50 mg/kg; however, no samples exceeded this level in the current study. In addition, the food additives regulations in Taiwan stipulate that the sulfite residue in mustard pickle products should not exceed 30 ppm. According to Kung et al. [2] and our results, the unqualified rates were 82.6% and 100%, respectively, indicating that sulfites over the limit standard are still used in mustard pickle products in Taiwan. Although added sulfites have bleaching and antibacterial effects on the product, their excessive addition can cause adverse reactions such as asthma and other allergic reactions in the human body.

### 3.4. HTS Analysis of Mustard Pickle Samples

The 16S rRNA gene sequences are commonly used to identify, quantify, and visualize microorganism populations in fermented foods [12,13] because their genes consist of highly conserved domains interspersed with variable regions. Comparative analysis of these sequences is a powerful means to infer phylogenetic relationships among organisms. The HTS technology is capable of analyzing the bacterial profiles of environmental samples based on the full length 16S rRNA gene. This study employed HTS technology for the full-length sequencing of the 16S rRNA genes of bacteria present in commercial mustard pickle samples. These analyses have enabled the community structures of the bacteria present in mustard pickle samples from the different markets to be compared for the first time.

Most of the PCR products of bacterial DNA from the samples were of poor quality. Therefore, only the PCR products from samples A, C, E, and J were subjected to HTS analysis. The relative abundances of bacterial species in samples A, C, E, and J are shown in Figure 2. The main phylum in sample A was Proteobacteria, accounting for 77%, followed by Firmicutes, accounting for 18%, whereas the remaining phylum was Bacteroidetes, accounting for 4%. The main phylum in sample C was Firmicutes, accounting for 89%, followed by Proteobacteria, accounting for 10%. The main phylum in sample E was Firmicutes, accounting for 99%. In sample J, the main phylum was Firmicutes, accounting for 71%, followed by Proteobacteria, accounting for 23%. The main bacterial class in sample A was Gammaproteobacteria, accounting for 76%, followed by Bacilli, accounting for 18%. The main bacterial class in sample C was Bacilli, accounting for 89%, whereas the remaining bacteria were Gammaproteobacteria, accounting for 5%, and Alphaproteobacteria, accounting for 4%. Bacilli was the main bacterial class in sample E, accounting for 99%. In sample J, Bacilli accounted for 69%, followed by Gammaproteobacteria, which accounted for 19%. The main order in sample A was Enterobacterales, accounting for 28%, followed by Vibrionales, accounting for 25%, whereas the remaining orders were Bacillales, accounting for 11%, and Moraxellales, accounting for 10%. The main order in sample C was Lactobacillales, accounting for 89%. Lactobacillales was the main bacterial order in sample E and accounted for 99%. In sample J, the main order was Lactobacillales, accounting for 68%. At the family level, the main bacterial family in sample A was *Morganellaceae,* accounting for 28%, followed by *Vibrionaceae,* accounting for 25%, and the other bacterial family was *Moraxellaceae,* accounting for 10%. The main bacterial family in samples C, E, and J was *Lactobacillaceae*, which accounted for 88%, 98%, and 68%, respectively. At the genus level, the main genera in sample A were *Vibrio* and *Proteus*, each accounting for 25%, whereas the other genera included *Psychrobacter,* accounting for 10%. The main genus in sample C was *Weissella,* accounting for 62%, followed by *Lactobacillus,* accounting for 15%. The main genus in sample E was *Lactobacillus,* accounting for 97%. The main genus in sample J was *Companilactobacillus,* accounting for 57%.

The cluster heat map of the species abundance of microorganisms is shown in Figure 3. The OTU data of the top 35 most abundant genera were selected to draw the heat map, which were presented as two-dimensional data, where the columns represented the sample groups and rows represented the OTUs. Genera showing high proportions in sample A included *Vibrio*, *Oceanimonas*, *Shewanella*, *Kurthia*, *Idiomarina*, *Psychrobacter*, *Proteus*, *Mesonia*, *Enterococcus*, *Zunongwangia*, and *Providencia*. Genera with high proportions in sample C included *Lactococcus*, *Methylobacterium*, *Leuconostoc*, *Weissella*, *Methylorubrum*, and *Latilactobacillus*. *Lactobacillus* and *Pluralibacter* showed high proportions in sample E. Genera with high proportions in sample J included *Acinetobacter*, *Pediococcus*, *Enterobacter*, *Pseudomonas*, *Paracoccus*, *Companilactobacillus*, *Anoxynatronum*, *Ectothiorhodospira*, *Citrobacter*, *Cutibacterium*, *Lentimicrobium*, *Aeromonas*, *Lactiplantibacillus*, and *Levilactobacillus*.

Based on the above results, the microbiomes of mustard pickle samples A, C, E, and J largely differed from each other. The main bacterial genus in sample A was *Proteus,* accounting for 25%; *Proteus* are Gram-negative bacteria that mainly exist in soil, water, and the intestinal tract of mammals (including humans). Some strains may cause urinary tract infections in humans [26,27], whereas other strains such as *Proteus mirabilis* can elicit gastroenteritis during food poisoning [28]. In addition, the genus *Vibrio* in sample A was dominant and accounted for 25%. These Gram-negative bacteria mainly grow in estuaries, seawater, and marine animals and include salt-tolerant marine bacteria [29]. Among them, some strains such as *Vibrio parahaemolyticus* are important pathogens that cause food poisoning [30]. Sample A may have been contaminated during storage or sales, such as through cross-contamination by products sold together in traditional markets, including seafood, meat products, and soy products, which may lead to spoilage and decay of mustard pickle products. The main bacterial genus in sample C was *Weissella,* which accounted for 62%. *Weissella* belongs to *Lactobacillaceae*, a family of Gram-positive bacteria, and is a relatively new member of the *Lactobacillus* family and consists of 23 species, including *W. kimchii*, *W. koreensis*, and *W. cibaria* [31]. In addition, the dominant genus in sample E was *Lactobacillus,* accounting for 97%, which also belongs to the Gram-positive bacteria family *Lactobacillaceae*. However, the dominant genus in sample J was *Companilactobacillus,* accounting for 57%, which also belongs to the Gram-positive family *Lactobacillaceae*. In summary, the microbiomes in different mustard pickle samples differed. The dominant bacteria in all samples except for in sample A was *Lactobacillaceae*. In contrast, the microbiome of sample A mainly consisted of *Morganellaceae* and *Vibrionaceae*, possibly because of environmental contamination during storage and sale.

Chao et al. [32] found that bacteria in *Lactobacillus,* such as *Lactobacillus*, *Pediococcus*, *Weissella*, and *Leuconostoc,* were important microbial flora in the fermentation of mustard pickle products in Taiwan. In Kimchi products, the dominant bacterial genera were *Leuconostoc*, *Lactobacillus*, and *Weissella* [33,34]. In addition, a study of Sichuan pickle brine and Chongqing radish pickle brine employing SMRT method showed that *Lactobacillus* was dominant [12,13]. Based on this polyphasic approach, Zheng et al. [35] proposed reclassification of the genus *Lactobacillus* into 25 genera including the emended genus *Lactobacillus* in 2020. Differences in the fermentation process, raw materials, and geographic distribution of fermented vegetables may considerably impact the bacterial composition of the products.

This study found that mustard pickle samples contained opportunistic pathogens such as *Proteus* spp. and *Vibrio* spp. The findings of this work were similar to those of previous research on other fermented vegetables [12,13,36]. For example, Yang et al. [13] recently showed that the bacterial composition of Chongqing radish paocai brines included *Pseudomonas* spp. and three opportunistic pathogens using PacBio SMRT analysis. Similarly, Cao et al. [12] found that certain opportunistic pathogens were also detected in home-made Sichuan pickles (Sichuan paocai) using PacBio SMRT analysis. In addition, Luo et al. [36] reported that the *Pseudomonas* and *Bacillus* genera were detected in Sichuan pickle products. Therefore, the above studies show that fermented vegetable products may contain opportunistic pathogenic bacteria in addition to lactic acid bacteria. It is reasonable to assume that there is safety risk in consuming fermented vegetables.

## 4. Conclusions

Our results revealed the favorable microbial quality of commercially available mustard pickle products, which contained no food-borne pathogenic bacteria or hygienic indicator bacteria. The biogenic amine contents of the samples were low (<48.0 mg/kg). However, the sulfite contents of all commercially available mustard pickle products were above 100 ppm, exceeding the limit set by food additive regulations (30 ppm), with a failure rate of 100%. Sulfite levels in samples above standard limits may cause asthma and other allergic reactions in consumers. Therefore, it is an allergen that poses a health hazard to humans. The microbiome of sample A mainly consisted of *Morganellaceae* and *Vibrionaceae*, whereas those of other samples mainly consisted of *Lactobacillaceae*. Sample A may have been contaminated by the environment during storage or sales, which can lead to product spoilage and even food poisoning. Therefore, the results of this study point to the need to take appropriate measures against high sulfite levels and potential sources of bacterial contamination to avoid risks to public health.

## Figures and Tables

**Figure 1 biology-12-00258-f001:**
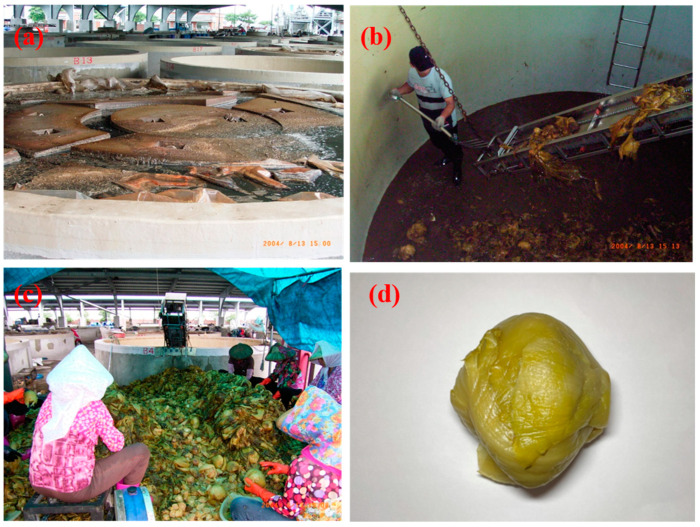
The photos of fermentation, selection, and product in mustard pickle: (**a**) Fermenting bucket with heavy stones on the cover; (**b**) The inside of the fermenting bucket; (**c**) Mustard pickle was taken out and selected; (**d**) Mustard pickle product.

**Figure 2 biology-12-00258-f002:**
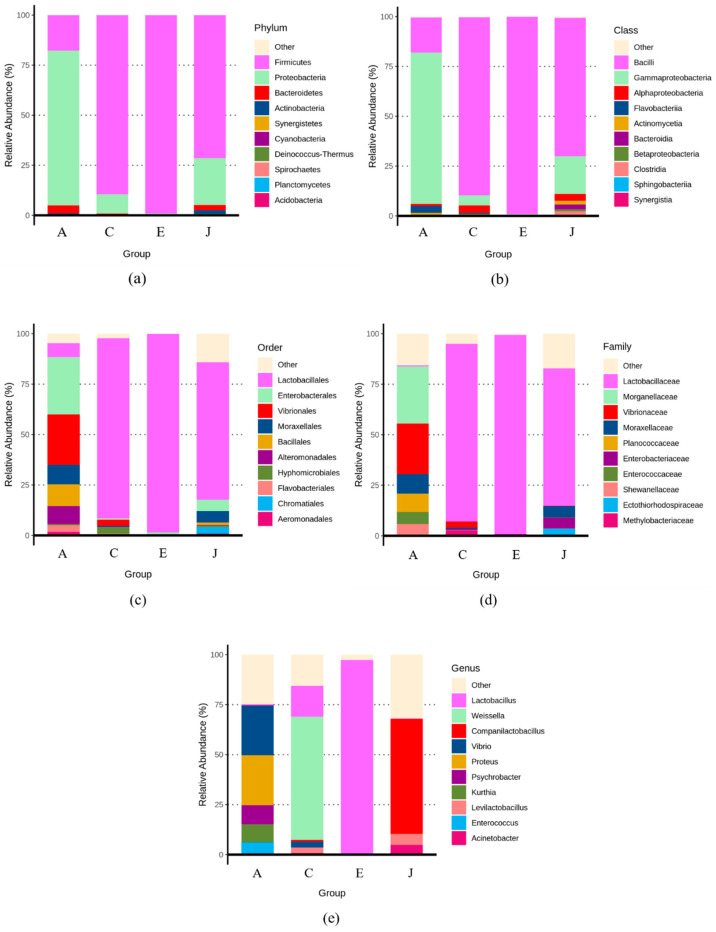
Bacteria species annotation of A, C, E, and J mustard pickle products in retail market on levels of phylum (**a**), class (**b**), order (**c**), family (**d**), and genus (**e**).

**Figure 3 biology-12-00258-f003:**
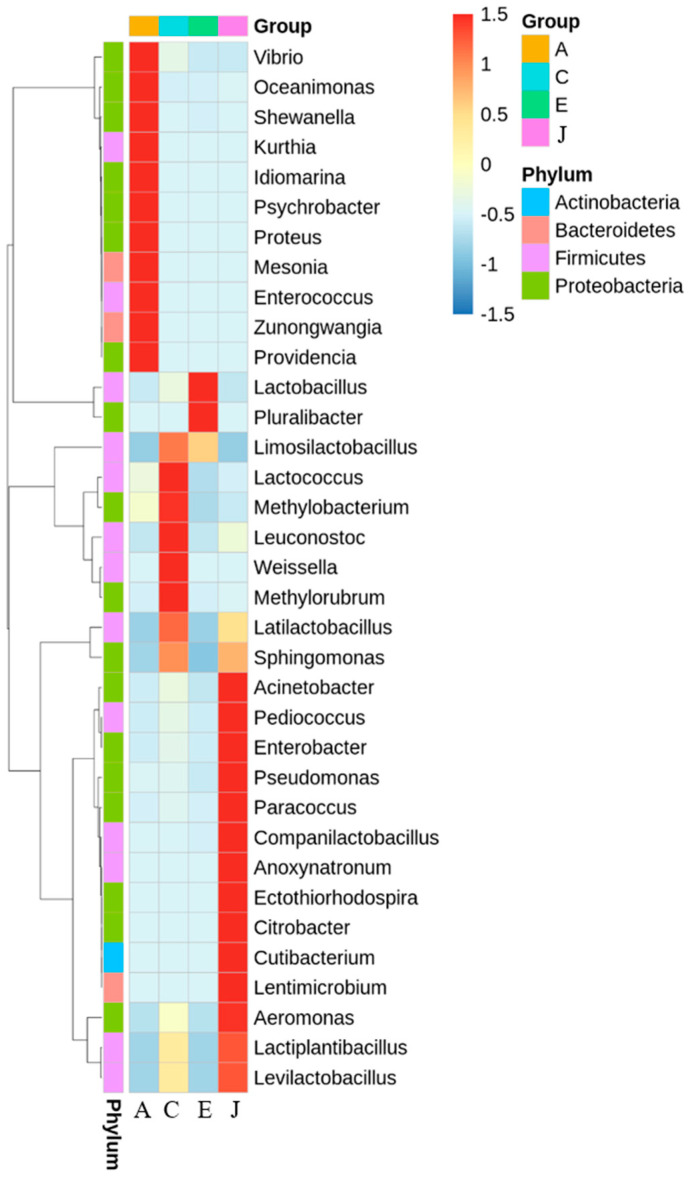
Heat map of microorganism species abundance of A, C, E, and J samples in retail market on genus level.

**Table 1 biology-12-00258-t001:** Aerobic plate count (APC), lactic acid bacteria count (LAB), coliform, *Escherichia coli*, yeast and mold, and pathogens (*Staphylococcus aureus*, *Salmonella* spp., *Listeria monocytogenes*) in mustard pickle products from retail market.

Sample Code	APC (log CFU/g)	LAB (log CFU/g)	Coliform (log CFU/g)	*Escherichia coli* (log CFU/g)	Yeast and Mold (log CFU/g)	*Staphylococcus aureus* (log CFU/g)	*Salmonella* spp. (log CFU/g)	*Listeria monocytogenes* (log CFU/g)
A	4.01 ± 0.07	3.30 ± 0.10	<1	<1	<1	<2	<2	<2
B	2.65 ± 0.05	2.00 ± 0.05	<1	<1	<1	<2	<2	<2
C	3.99 ± 0.05	3.44 ± 0.07	<1	<1	<1	<2	<2	<2
D	3.18 ± 0.05	<1	<1	<1	<1	<2	<2	<2
E	3.73 ± 0.12	3.77 ± 0.11	<1	<1	4.85 ± 0.07	<2	<2	<2
F	2.98 ± 0.13	2.47 ± 0.07	<1	<1	<1	<2	<2	<2
G	2.59 ± 0.19	<1	<1	<1	<1	<2	<2	<2
H	2.18 ± 0.05	<1	<1	<1	<1	<2	<2	<2
I	2.91 ± 0.07	<1	<1	<1	2.68 ± 0.20	<2	<2	<2
J	3.34 ± 0.05	2.42 ± 0.12	<1	<1	2.30 ± 0.60	<2	<2	<2
K	2.40 ± 0.04	<1	<1	<1	<1	<2	<2	<2
L	2.63 ± 0.15	<1	<1	<1	<1	<2	<2	<2
M	3.13 ± 0.06	<1	<1	<1	<1	<2	<2	<2
N	2.36 ± 0.18	<1	<1	<1	<1	<2	<2	<2
Range	2.18–4.01	<1–3.77	<1	<1	<1–4.85	<2	<2	<2
Average	3.01 ± 0.59 ^a^	1.24 ± 1.55	<1	<1	0.70 ± 1.50	<2	<2	<2

^a^ Means ± S.D.

**Table 2 biology-12-00258-t002:** pH, water activity (Aw), moisture, salt content, titratable acidity, total volatile basic nitrogen (TVBN), sulfite content, and nitrite content in mustard pickle products from retail market.

Sample Code	pH	Aw	Moisture (%)	Salt Content (%)	Titratable Acidity (%)	TVBN (mg/100 g)	Sulfite Content (ppm)	Nitrite Content (ppm)
A	4.16 ± 0.08	0.951 ± 0.06	90.20 ± 0.55	3.51 ± 0.10	0.51 ± 0.04	19.5 ± 0.5	237.6 ± 4.9	0.40 ± 0.02
B	3.39 ± 0.03	0.950 ± 0.08	91.24 ± 0.60	3.50 ± 0.60	1.66 ± 0.16	19.9 ± 0.6	274.6 ± 8.8	<0.01
C	3.46 ± 0.03	0.945 ± 0.03	89.02 ± 0.40	3.89 ± 0.13	1.35 ± 0.10	20.0 ± 0.8	337.0 ± 14.7	0.12 ± 0.01
D	3.84 ± 0.08	0.948 ± 0.04	89.44 ± 0.31	4.07 ± 0.70	0.78 ± 0.05	21.3 ± 0.6	101.3 ± 4.3	0.27 ± 0.06
E	3.67 ± 0.03	0.945 ± 0.02	89.74 ± 0.41	4.31 ± 0.16	1.13 ± 0.04	33.3 ± 0.7	195.3 ± 6.8	0.50 ± 0.21
F	3.96 ± 0.02	0.947 ± 0.03	89.65 ± 0.71	4.10 ± 0.20	0.73 ± 0.06	22.3 ± 0.6	837.3 ± 8.4	0.34 ± 0.10
G	4.07 ± 0.03	0.948 ± 0.05	89.73 ± 0.52	5.32 ± 0.14	0.67 ± 0.06	16.6 ± 0.5	1065.4 ± 14.9	0.40 ± 0.06
H	3.90 ± 0.02	0.947 ± 0.06	89.58 ± 0.90	5.52 ± 0.40	0.74 ± 0.07	23.7 ± 0.5	144.0 ± 5.7	0.59 ± 0.04
I	4.04 ± 0.06	0.921 ± 0.05	87.59 ± 0.41	8.48 ± 0.30	0.64 ± 0.07	19.6 ± 0.7	411.3 ± 10.9	0.65 ± 0.09
J	4.08 ± 0.07	0.945 ± 0.04	89.96 ± 0.90	4.86 ± 0.80	0.60 ± 0.05	21.9 ± 0.5	473.2 ± 10.6	0.09 ± 0.01
K	3.59 ± 0.06	0.914 ± 0.04	85.77 ± 0.40	8.55 ± 0.10	1.11 ± 0.12	36.0 ± 0.9	246.7 ± 4.9	0.37 ± 0.05
L	3.87 ± 0.04	0.960 ± 0.06	92.18 ± 0.30	3.02 ± 0.70	0.64 ± 0.07	17.7 ± 0.6	753.7 ± 6.6	0.31 ± 0.04
M	3.84 ± 0.03	0.954 ± 0.04	92.80 ± 0.61	3.12 ± 0.40	0.72 ± 0.06	18.3 ± 0.6	787.2 ± 17.1	0.19 ± 0.05
N	3.61 ± 0.05	0.943 ± 0.06	88.88 ± 0.70	4.23 ± 0.19	0.98 ± 0.05	22.6 ± 0.7	301.9 ± 11.7	0.24 ± 0.03
Range	3.39–4.16	0.914–0.960	85.77–92.80	3.02–8.55	0.51–1.66	16.6–36.0	101.3–1065.4	<0.01–0.65
Average	3.82 ± 0.24 ^a^	0.944 ± 0.012	89.70 ± 1.74	4.75 ± 1.76	0.88 ± 0.33	22.3 ± 5.6	440.5 ± 299.7	0.32 ± 0.19

^a^ Means ± S.D.

**Table 3 biology-12-00258-t003:** Level of biogenic amines in mustard pickle products from retail market.

Sample Code	Levels of Biogenic Amine (mg/kg)								
	Try ^a^	Phe	Put	Cad	His	Tyr	Spd	Spm	Total
A	0.5 ± 0.1	7.7 ± 0.3	3.1 ± 0.1	19.6 ± 0.9	1.7 ± 0.5	29.3 ± 4.3	6.2 ± 0.7	ND ^c^	68.1
B	1.7 ± 0.4	5.5 ± 0.1	5.0 ± 0.2	16.6 ± 1.1	3.8 ± 0.6	40.5 ± 3.9	4.7 ± 0.4	1.0 ± 0.3	78.8
C	3.2 ± 0.6	6.2 ± 0.6	4.6 ± 0.3	17.8 ± 1.3	5.4 ± 0.5	42.9 ± 5.1	5.7 ± 0.6	ND	85.8
D	0.5 ± 0.1	7.2 ± 0.2	3.1 ± 0.1	8.6 ± 0.8	1.6 ± 0.3	14.7 ± 1.8	7.7 ± 0.7	ND	43.4
E	6.7 ± 0.5	7.7 ± 0.5	5.8 ± 0.5	25.1 ± 5.5	2.4 ± 0.2	73.4 ± 8.2	9.5 ± 0.8	1.6 ± 0.6	132.2
F	3.8 ± 0.3	6.7 ± 0.1	3.4 ± 0.5	16.9 ± 2.1	3.1 ± 0.7	26.1 ± 6.0	6.7 ± 0.5	ND	66.7
G	3.5 ± 0.2	5.8 ± 0.1	4.7 ± 0.6	7.3 ± 0.8	1.3 ± 0.4	56.9 ± 6.2	7.0 ± 0.6	ND	86.5
H	4.1 ± 0.2	8.7 ± 0.4	4.0 ± 0.2	50.1 ± 7.3	7.6 ± 0.6	32.8 ± 5.1	7.8 ± 0.7	ND	115.1
I	0.5 ± 0.1	17.9 ± 0.9	5.3 ± 0.1	9.1 ± 0.7	11.0 ± 0.5	19.9 ± 3.7	4.3 ± 0.2	ND	68.0
J	0.5 ± 0.1	13.3 ± 0.7	6.0 ± 0.5	27.2 ± 3.0	17.3 ± 2.9	38.0 ± 5.0	4.9 ± 0.3	ND	107.2
K	3.4 ± 0.5	12.7 ± 0.7	6.8 ± 0.4	31.9 ± 4.1	8.6 ± 1.6	64.7 ± 9.2	6.8 ± 0.7	0.8 ± 0.4	135.7
L	0.5 ± 0.2	8.9 ± 0.6	4.5 ± 0.1	20.9 ± 1.6	6.6 ± 1.1	34.2 ± 5.1	3.6 ± 0.5	ND	79.2
M	0.5 ± 0.1	8.9 ± 0.6	5.2 ± 0.3	26.2 ± 5.0	5.6 ± 0.9	43.2 ± 7.3	4.5 ± 0.6	ND	94.1
N	0.5 ± 0.3	10.6 ± 0.8	6.1 ± 0.6	39.2 ± 3.8	14.0 ± 2.8	55.0 ± 6.8	4.5 ± 0.4	0.6 ± 0.1	130.5
Range	0.5–6.7	5.5–17.9	3.1–6.8	7.3–50.1	1.3–17.3	14.7–73.4	3.6–9.5	ND–1.6	43.4–135.7
Average	2.1 ± 0.2 ^b^	9.1 ± 0.4	4.8 ± 0.1	22.6 ± 1.2	6.4 ± 0.5	48.0 ± 1.7	6.0 ± 0.2	0.3 ± 0.1	78.7 ± 65.1

^a^ Try: tryptamine; Phe: 2-Phenylethylamine; Put: Putrescine; Cad: Cadaverine; His: Histamine; Tyr: Tyramine; Spd: Spermidine; and Spm: Spermine. ^b^ Means ± S.D. ^c^ ND, Not detected (amine level less than 0.05 mg/kg).

**Table 4 biology-12-00258-t004:** Correlation coefficients among aerobic plate count (APC), lactic acid bacteria count (LAB), yeast and mold, pH, water activity (Aw), moisture, salt content, titratable acidity, total volatile basic nitrogen (TVBN), and sulfite and nitrite contents in mustard pickle products from retail market.

	APC	LAB	Yeast and Mold	pH	Aw	Moisture	Salt Content	Titratable Acidity	TVBN	Sulfite	Nitrite
APC											
LAB	0.7913 **										
Yeast and mold	0.3489	0.3857									
pH	0.0998	−0.1165	0.0980								
Aw	0.2096	0.2025	−0.2363	0.1137							
Moisture	0.1493	0.0873	−0.1429	0.1092	0.9020 **						
Salt content	−0.3527	−0.3536	0.2379	0.1009	−0.9384 **	−0.8417 **					
Titratable acidity	−0.0080	0.2902	−0.0051	−0.9390 **	−0.1248	−0.1197	−0.0693				
TVBN	−0.0518	0.1619	0.4139	−0.3597	−0.5750*	−0.5921 *	0.4538	0.32020			
Sulfites	−0.1568	−0.2023	−0.2045	0.3786	0.2915	0.3860	−0.1637	−0.36741	−0.4862		
Nitrite	−0.1651	−0.2344	0.3410	0.4588	−0.3764	−0.3888	0.5487*	−0.46315	0.2469	−0.0767	

* Significant correlation (*p* < 0.05); ** Extremely significant correlation (*p* < 0.01).

## Data Availability

Not applicable.

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
