# Peer review of "Determination of the Bacterial Community of Mustard Pickle Products and Their Microbial and Chemical Qualities"

_biology, 2023, doi:10.3390/biology12020258_

Round 1

Reviewer 1 Report (Previous Reviewer 2)

The article was much improved. I have only some comments.

·       What I find missing from the introduction is a brief paragraph on the contaminants detected in the tested material (by other authors), and which the authors also examine in the article.

·       All abbreviations should be defined when used for the first time.

·       There are some typos in the manuscript.

·       In Figures 2 and 3 the microorganisms names should be written in italics.

Author Response

Reviewer 2 Report (Previous Reviewer 3)

Authors have satisfactorily responded to all my questions and made the necessary changes.

Author Response

Thanks for the review’s comment. We appreciate the reviewers’ efforts in providing valuable comments for us to improve the quality of the manuscripts.

This manuscript is a resubmission of an earlier submission. The following is a list of the peer review reports and author responses from that submission.

Round 1

Reviewer 1 Report

Dear authors,

The topic is important due to the enlightening of the sulfite content (from bleach?) that was exceeding the limits in the products.

1. The title is too long.

2. The abstract is messy. Try to avoid  punctuating the different results from for example the samples (A,C,E and J).  The abstract should give a good overview of the total work. When using the genera Lactobacillus, please be sure of you use the correct name as the genera Lactobacillus has recently been reclassified into 25 genera (Zheng et al (2020) International Journal of systematic and evolutionary microbiology, 70, 2782-2858.

3. Introduction. The plant "mustard" must be named also in latin as there are several "mustards. Nice with pictures! If you investigate an improvement during the last ten years, what are you comparing with? More information is needed.

4. Method. Please motivate why you compare the number of microorganisms with results from chemical analysis. I am not convinced. Also when applying the statistical analysis Pearson´s correlation coefficient analysis- why using that? Please motivate that choice.

5. By using the HTS analysis, the results were presented at phylum level. What does it tell the readers? I have got a feeling of that you applied HTS analysis just per se to demonstrate it. But please be more clear in your description on why you needed to use HTS?? In 3.4. The text is not in connection with what you have been doing elsewhere.

6. Also in your discussion when you refer to Chao et al (2009); Jeong et al ((2013); Jung et al (2011) you should emphasize that when it comes to the genera Lactobacillus there are now 25 different genera of the past Lactobacillus genus.

7. Conclusion    The lines 342-347 seems to be interesting and OK, however, what happens if you cut off just the HTS analysis (what is SMRT??)? For me it seems like if it unnecessary. If I do not understand why you use the HTS it might depend on that you did not motivate the reader enough for it. What aim did you use for that?

8. There are some major changes to be done- Good luck!

Reviewer 2 Report

·       The title is too long and should be shortened.

·       The name of the microbial family should be written in italics.

·       Line 45: “In traditional Taiwanese markets, mustard…” finish the sentence please.

·       What I find missing from the introduction is a brief paragraph on the contaminants detected in the tested material (by other authors), and which the authors also examine in the article.

·       What is the scientific value of the research? The article sounds like a kind of a report.

·       Emphasise the importance of the study please. Maybe in the aspect of public health.

Reviewer 3 Report

This manuscript by Chien  et al. describes “Microbial and Chemical Qualities and Bacterial Community in Mustard 2 Pickle Products, a Traditional Fermented Vegetable in Taiwan,

Determined Using High-throughput Sequencing.  However, the manuscript is in many places poorly written. There are also important gaps in documentation of the method, missing references, etc.

Line 22:  “no coliform ??“ no coliform bacteria

Line 23:  “and” to or

Line 28: “ Proteus (25%), Vibrio (25%)..etc”  to Proteus spp

Line 32:  “except for in sample A” to sample A

Author only used 8 references in entire introduction. Please add a reference to following lines (Line 39,41, 52-55,65..etc).

Line 67: rewrite this paragraph

Line 80-84: missing references

Line 108-109: “de Man,  Rogosa and Sharpe agar”  add (MRS agar) ??

Line 100-117: missing references

Discussion section: Author has not discussed his findings at all. Only described his results.

The references need some work – there are some odd capitals/lacks of capitals. Please check the relevant section in the instructions for authors for more details

English grammar requires improvement throughout the manuscript.

Round 2

Reviewer 1 Report

Hi!

It seems like if the authors have made the necessary improvements.

Author Response

Response 1Thanks for the review’s comment. We appreciate the reviewers’ efforts in providing valuable comments for us to improve the quality of the manuscripts.

Reviewer 3 Report

The manuscript has improved significantly, and the authors have satisfactorily responded to all my questions and made the necessary changes. However, there remain some corrections that I believe are necessary to be addressed. 

Line 25: please remove (100%, 14/14)

Line 209-212:  Author mentioned that sulfites could kill or inhibit the bacteria in mustard pickle. Please explain that with some references.

Line 231-232:  {………such as asthma in consumers} please add reference.
